# Laboratory and Field Assessments of Oral *Vibrio* Vaccine Indicate the Potential for Protection against Vibriosis in Cultured Marine Fishes

**DOI:** 10.3390/ani12020133

**Published:** 2022-01-07

**Authors:** Aslah Mohamad, Fathin-Amirah Mursidi, Mohd Zamri-Saad, Mohammad Noor Azmai Amal, Salleh Annas, Md Shirajum Monir, Mohd Loqman, Fahmie Hairudin, Nurhidayu Al-saari, Md Yasin Ina-Salwany

**Affiliations:** 1Aquatic Animal Health and Therapeutics Laboratory, Institute of Bioscience, Universiti Putra Malaysia, Serdang 43400, Selangor, Malaysia; aslahumt@gmail.com (A.M.); fathimi91@yahoo.com (F.-A.M.); mzamri@upm.edu.my (M.Z.-S.); mnamal@upm.edu.my (M.N.A.A.); annas@upm.edu.my (S.A.); 2Department of Veterinary Laboratory Diagnosis, Faculty of Veterinary Medicine, Universiti Putra Malaysia, Serdang 43400, Selangor, Malaysia; 3Department of Biology, Faculty of Science, Universiti Putra Malaysia, Serdang 43400, Selangor, Malaysia; 4Department of Aquaculture, Faculty of Agriculture, Universiti Putra Malaysia, Serdang 43400, Selangor, Malaysia; monir_bau22@yahoo.com (M.S.M.); Loqman93@gmail.com (M.L.); muhamad_fahmie23@yahoo.com (F.H.); 5International Institute for Halal Research and Training (INHART), Level 3, KICT Building, International Islamic University Malaysia (IIUM), Gombak 53100, Selangor, Malaysia; hidayusaari@iium.edu.my

**Keywords:** vibriosis, *Vibrio harveyi*, oral vaccine, marine fishes

## Abstract

**Simple Summary:**

The application of chemicals and drugs in fish culture has increased the existence of antimicrobial-resistant bacteria and drug residues in fish. Developing an effective vaccine to reduce specific disease-related losses and avoid antibiotic use has become an increasingly important part of aquaculture. Our current study found that the oral inactivated *V. harveyi* strain VH1 vaccine confers high protection in fish post-infection with virulent *Vibrio* spp. under experimental conditions. The oral vaccine was further tested for field efficacy and showed a higher survival and growth performance in vaccinated fish than unvaccinated fish. This current vaccine was shown to potentially provide sufficient protection to the host in both controlled and field environments against vibriosis.

**Abstract:**

Vibriosis is one of the most common threats to farmed grouper; thus, substantial efforts are underway to control the disease. This study presents an oral vaccination against multiple *Vibrio* spp. in a marine fish with double booster immunisation. The *Vibrio harveyi* strain VH1 vaccine candidate was selected from infected groupers *Epinephelus* sp. in a local farm and was formalin inactivated and combined with commercial feed at a 10% ratio (*v*/*w*). A laboratory vaccination trial was conducted for seventy days. The induction of IgM antibody responses in the serum of Asian seabass *Lates calcarifer* immunised with the oral *Vibrio harveyi* strain VH1 was significantly (*p* < 0.05) increased as early as week one post-primary vaccination. Subsequent administration of the first and second booster for 5 consecutive days, starting on days 14 and 42, respectively, improved the specific antibody level and reached a highly significant (*p* < 0.05) value at days 35 and 49 before slightly decreasing from day 56 onwards. Antibody titres of the control unvaccinated group remained relatively stable and low throughout the experimental period. At the end of the 70-day vaccination trial, 23 days post final boost, an intraperitoneal challenge with a field strain of *Vibrio harveyi*, *V. alginolyticus,* and *V. parahaemolyticus* was carried out. Our challenge study showed that oral *Vibrio harveyi* strain VH1 vaccine candidate could induce significant protection, with an RPS of 70–80% against different *Vibrio* species. Thereafter, a field trial was conducted in a mariculture farm to study the effect of field vaccination using the oral *Vibrio harveyi* strain VH1 vaccine candidate. A total of 3000 hybrid grouper juveniles were divided into two groups in triplicate. Fish of Group 1 were not vaccinated, while Group 2 were vaccinated with the feed-based vaccine. Vaccinations were carried out on days 0, 14, and 42 via feeding the fish with the vaccine at 4% body weight for 5 consecutive days. At the end of the study period, the fish survival rate was 80% for the vaccinated group, significantly (*p* < 0.05) higher than the 65% seen in the control unvaccinated group. Furthermore, the vaccinated fish showed significantly (*p* < 0.05) better growth performances. Therefore, the oral *Vibrio* vaccine from the inactivated *Vibrio harveyi* strain VH1 is a potential versatile vaccine candidate that could stimulate good immune responses and confer high protection in both Asian seabass, *Lates calcarifer,* and farm hybrid grouper *Epinephelus fuscoguttatus* × *Epinephelus lanceolatus.*

## 1. Introduction

In Malaysia and its neighbouring countries, with a year-round tropical climate of 28 °C, vibriosis has been frequently recorded at many marine aquaculture farms. In many outbreaks, *Vibrio harveyi*, *V. parahaemolyticus*, *V. alginolyticus,* and *V. anguillarum* were most frequently isolated, affecting Asian seabass *Lates calcarifer*, brown-marbled grouper *Epinephelus fuscoguttatus*, orange-spotted grouper *Epinephelus coioides*, snappers *Lutjanus* sp., and hybrid grouper (brown-marbled grouper × giant grouper *E. lanceolatus*) [1,2,3,4]. Mohamad et al. [5] reported an outbreak in Selangor, Malaysia, where farmed juvenile hybrid grouper (*E. polyphekadion × E. fuscoguttatus*) experienced losses of 29% in 10 days, with diseased fish becoming lethargic, displaying excessive mucus production, fin rot, congestion of the brain, liver, and kidneys, and splenic enlargement. *Vibrio harveyi* and *V. alginolyticus* were successfully recovered from the diseased hybrid groupers and may cause the infection of the fish host. Another study by Amalina et al. [6] on groupers obtained from nine farms located at different geographical regions in Malaysia had detected multiple *Vibrio* spp. from 72% of the sampled grouper. *Vibrio communis* showed the highest prevalence in grouper, followed by *V. parahaemolyticus, V. alginolyticus, V. vulnificus, V. rotiferianus, V. campbellii, V. mytili, V. furnissii, V. harveyi, V. tubiashii, V. fluvialis,* and *V. diabolicus*. In China, Large yellow croaker, *Pseudosciaena crocea*, an economically important fish species of mariculture, was commonly isolated with multiple *Vibrio* spp. such as *Vibrio harveyi*, *Vibrio alginolyticus,* and *Vibrio parahaemolyticus* [7]. More than one pathogen is usually isolated from the sick fish [8] and causes more severe disease than a single bacterial infection [9]. The incidence creates an urgent need to develop versatile or combined vaccines to simultaneously control fish disease caused by multiple pathogens [8].

Vaccination is one of the alternatives proposed to overcome the disease-caused mortality and morbidity after the restriction of using antibiotics in aquaculture because vaccines are more effective and safer than antibiotics to humans and the environment [1]. As opposed to antibiotics that aim to kill or stop diseases, vaccines, on the other hand, stimulate the fish’s immune system for antibody production, thus effectively managing fish diseases [10]. Although vibriosis can be controlled through vaccination [11], the existence of different strains and antigenic diversities of *Vibrio* species and their serotypes have led to slow progress of vaccine development [12,13]. Therefore, developing a versatile vaccine that can fight against multiple *Vibrio* by eliciting protection against homologous and heterologous strains is urgently needed to hinder vibriosis infections [14].

Developing a versatile vaccine that can be used to treat multiple infections will simultaneously provide practical ease of application while decreasing workload compared with the other ways of vaccination [15]. Economically, countering multiple *Vibrio* spp. using one application is cost-effective rather than buying a separate vaccine [16]. The critical part of developing an ideal and effective vaccine is identifying suitable antigens and important immunogenic sites [17]. In aquaculture, oral vaccination was described as a feasible immunisation method for fish farmers as there is no specific technical skill required to administer the vaccine and no direct interaction between handler and fish [18]. Moreover, oral vaccine administration provides an easier, cost-effective, user-friendly, and less stressful vaccination method [19].

Our previous study on a feed-based whole-cell polyvalent vaccine against vibriosis, streptococcosis, and motile aeromonad septicaemia in Asian Seabass, *Lates calcarifer,* showed that the oral polyvalent vaccine could provide around 75–80% protection after challenge with *V. harveyi*, *A. hydrophila*, and *S. agalactiae* [20]. The *Vibrio harveyi* strain VH1 vaccine, which was a part of the polyvalent vaccine, was still not tested as a single oral vaccine against different *Vibrio* spp.; thus, it was used in the current study to determine its ability to develop an antibody response shared against other *Vibrio* pathogens. Therefore, this research study proposes an oral *Vibrio harveyi* strain VH1 vaccine candidate that can provide good protection under laboratory, i.p. administrated, challenge models against three major *Vibrio* species; *V. harveyi*, *V. parahaemolyticus,* and *V. alginolyticus,* and can potentially improve antibody response, survival, and growth performance of farm marine fish.

## 2. Materials and Methods

### 2.1. Bacterial Strain Selection

The bacterial strains used in this study are shown in Table 1. The pathogenic *Vibrio* spp. was previously isolated upon infection with vibriosis in deep-sea cages in Langkawi, Malaysia. Identification of this strain was made using 16SrRNA analysis. A total of 10 µL of each strain was cultured on Difco thiosulfate–citrate–bile salts–sucrose (TCBS) (Thermo Scientific, Waltham, MA, USA) agar plate for 18 h at 30 °C followed with incubation in 250 mL of trypticase soy broth (TSB) (Merck, Germany) at 30 °C for 24 h under 200 rpm in incubator shaker TSI-45 (Tech-Lab Scientific, Malaysia). Then, the stocks were stored in TSB supplemented with 1.5% NaCl and 20% glycerol at −80 °C for further use.

#### Preparation of Formalin-Killed Cells (FKC) of Vibrio Harveyi Strain VH1

*Vibrio harveyi* strain VH1 was grown on TSB broth supplemented with 1.5% NaCl followed with incubation in a shaker incubator at 30 °C and 150 rpm for 16 h. A serial dilution and standard plate count techniques were used to determine the bacteria concentration [21]. Briefly, 0.1 mL from the highest dilution was poured and spread onto the TCBS agar and incubated at 30 °C for 18 h. Between 25 to 250 colonies were counted before the concentration was expressed as colony-forming unit per millilitre (CFU/mL) according to the following equation:$$\frac{CFU}{mL}=\frac{\left(Number\;of\;colonies\;\times \;dilution\;factor\right)}{Volume\;of\;culture\;plate}$$

The bacteria culture was inactivated by adding 0.5% formalin (*v*/*v*) to the culture, followed by a 24 h incubation at 4 °C. The inactivated bacteria were centrifuged at 3000× *g* for 10 min, washed three times with sterile PBS to remove the formalin, adjusted at 1 × 10^8^ CFU/mL in PBS, and stored at 4 °C.

### 2.2. Preparation of Feed Vaccine

In this study, formalin-killed cells (FKCs) of *V. harveyi* strain VH1 prepared earlier was used for the feed vaccine preparation. The inactivated *V. harveyi* strain VH1 cells were washed four times with sterile PBS by centrifugation at 6000× *g* for 15 min to remove the media and formalin residue from the culture. Afterward, the inactivated bacteria were resuspended in sterile PBS at a concentration of 6.7 × 10^7^ CFU/mL and was streaked again onto TSA supplemented with 1.5% NaCl and incubated at 30 °C overnight to confirm that all bacterial cells were inactivated.

The formalin-killed whole cells strain was later added with a 10% (*v*/*w*) ratio before the mixture was thoroughly mixed with pelleted feed (Star Feed, Star Feed Mills SDN. BHD, Klang, Malaysia) to provide a final concentration of 10^6^ CFU/g of feed. For control, only phosphate-buffered saline (PBS) and palm oil were mixed with the pelleted feed. The method and composition of the oral vaccine have been filed for a patent (MyIPO Malaysia, patent No.: PI2021000105). Nutrient proximate compositions analysis was conducted following Sulaiman et al. [22], and the compositions were found to not differ significantly from the original feed.

### 2.3. Fish Vaccination and Sample Collection in Laboratory Trials

The use of experimental animals in this study was approved by Animal Care and Use Committee Universiti Putra Malaysia (UPM/IACUC/AUP-R078/2019). At the start of the trial, 400 healthy juveniles of Asian seabass, *Lates calcarifer* (15.8 ± 2.6 g in weight), were randomly divided into two groups with a duplicate. Each group was reared in two fibre-glass tanks (~500-litre capacity) with 100 fish per tank, with a stocking density of 38 fish/m^3^. The fish were not fed for a day before the experiment to ensure maximum feed-vaccine uptake. Group 1 was given the control feed that contained PBS and palm oil, while Group 2 was fed with the feed-based vaccine for five consecutive days at 4% body weight. Single and double boosters were given to the vaccinated group on days 14 and 42 in the same manner. On the other days, all fish were fed with untreated commercial feed pellets (Star Feed, Star Feed Mills SDN. BHD, Klang, Malaysia) until the end of the experimental period. Before sample collection, the fish were anaesthetised at the dose of 50 mg/L with tricaine methanesulphonate (MS-222; Sigma Aldrich, St. Louis, MI, USA). Blood samples (approximately 300 µL) were collected lethally from six fish per group through the caudal vein every 7 days, put into a 1.5 mL tube, and kept at 4 °C for 24 h to separate serum from the red blood cells (RBC). Serum was obtained from the blood samples by centrifugation at 13,000 rpm for 10 min at 4 °C. The following water parameters were measured using a YSI Pro Plus multiparameter instrument (Yellow Springs Instrument, Yellow Springs, OH, USA) and maintained throughout the acclimatisation and experimental periods: temperature at 25.28 ± 0.81 °C; pH at 7.66 ± 0.06; salinity at 27.22 ± 0.78 ppt; dissolved oxygen at 5.93 ± 0.25 mg/L; a photoperiod of 12 h daylight and 12 h darkness. A brief experimental design and feeding regime are shown in Figure 1.

### 2.4. Determination of Specific Serum Antibody Production

Serum samples were subjected to indirect ELISA to determine the IgM level, according to Firdaus-Nawi et al. [23], with minor modifications. Flat-bottom microtitre plates were coated with 100 µL coating antigens containing 10^5^ CFU/mL *V. harveyi*, *V. alginolyticus,* and *V. parahaemolyticus* separately in carbonate–bicarbonate buffer per well. The plates were left overnight at 4 °C before two times washing them with phosphate-buffered saline +0.05% Tween 20 (PBST). Then, 200 µL of 1% bovine serum albumin (BSA) diluted in PBS was added, and the plates were incubated for 1 h at 37 °C. Next, after the reaction well was washed three times with PBST, 100 µL of 1:100 serum diluted in PBS were inserted into the reaction well and incubated again for 1 h at 37 °C. Unbound antibodies were removed by washing thrice with PBST. Specific IgM was detected using anti-Asian seabass IgM monoclonal antibody (Aquatic Diagnostics Ltd., Oban, UK, 1/33 in PBS, 1 h) followed by incubation with anti-mouse-HRP (1/5000, Nordic, 1 h). After three washes with PBST, 100 µL of TMB substrate solution (Thermo Fisher Scientific, Waltham, MA, USA) was added to the reaction well to detect the bound conjugate before the reaction was stopped with 0.2 mol/L sulphuric acids. Values were obtained by measuring the absorbance at 450 nm using a Multiskan spectrum microplate reader (Thermo Scientific, Vantaa, Finland).

### 2.5. Experimental Challenge of Vibrio sp.

For efficacy trials, 100 fish from each group were i.p. injected with either *V. harveyi* strain VH1 or *V. parahaemolyticus* strain VPK1 or *V. alginolyticus* strain VA2, combined with 3 bacteria (with equal concentration for each bacteria) by intraperitoneal injection with 10^7^ CFU bacteria/fish [24] and PBS for non-challenged control (*n* = 10 with duplicate). In all experiments, mortalities were monitored daily for seven days, and the cause of death was established by isolating the challenge strains from visceral organs using TCBS and incubating them at 28 °C for 24 h. Fish were starved 24 h before the challenge. On the day of the challenge, the fish were anaesthetised with Metacaine and i.p. injected with 0.1 mL of the challenge strain. No mortality or abnormal behaviour was observed associated with the challenge procedure. The relative percentage survival (RPS) of Asian seabass immunised with the oral *Vibrio harveyi* strain VH1 vaccine was calculated to evaluate the efficacy of vaccination as RPS = 1 − (mortalities of vaccinated fish/mortalities of control fish) × 100.

### 2.6. Field Trial

The field vaccination trial was carried out for 112 days. At the start of the trial, hybrid grouper *Epinephelus fuscoguttus* × *E. lanceolatus* was divided into 1500 fish/groups with triplicates. The fish were not fed for a day before the experiment to ensure maximum feed-vaccine uptake. Group 1 was given the control feed that contained PBS and palm oil, while Group 2 was fed with the feed-based vaccine for five consecutive days. Single and double boosters were given to the vaccinated group on days 14 and 42 in the same manner. The fish was fed with the vaccinated feed at 4% body weight, while other husbandry practices were maintained. Water quality parameters such as pH, temperature, salinity, dissolved oxygen, and ammonia–nitrogen were monitored using YSI Pro Plus (Yellow Spring Instrument, Yellow Spring, OH, USA) and spectrophotometer (HACH Company, Loveland, CO, USA) were observed weekly until the end of the 16-week experimental period. Fish mortalities and abnormal features were recorded, while the survival rate was calculated at the end of the 112-day study. Moreover, the total weight to nearest 0.1 g of 10 randomly sampled fish in each group was determined while kidney samples were collected for bacterial isolation at 2-week intervals. Feed conversion efficiency was calculated as follows:$$\frac{Total\;consumption\;of\;feed\;\left(g\;feed\right)}{\left(Total\;number\;of\;fish-dead\;fish\right)\;\times \;Average\;growth\;per\;fish\;\left(g\right)}$$

#### 2.6.1. Bacterial Isolation and Identification

Samples of kidney were cultured for *Vibrio* spp. on thiosulphate–citrate–bile–salts–sucrose (TCBS) (Oxoid, Hampshire, UK) agar and tryptone soy broth (TSB) (Oxoid), with the addition of NaCl (1.5% *w*/*v*) at 30 °C for 24 h. The dominant bacterial colonies were sub-cultured to obtain pure colonies. The isolates then proceeded to the Gram-staining procedure, PCR, and sequencing for identification. Genomic DNA of pure colonies was extracted using DNeasy Blood and Tissue kit (QIAGEN, Hilden, Germany) according to the manufacturer’s protocol. The genomic DNA was subjected to PCR amplification using gyrB primers (Table 2). The PCR reactions were performed using REDiant 2× PCR Master Mix (FirstBase, Kuala Lumpur, Malaysia) in a final volume of 25 μL containing 2× PCR master mix, 1 µM of each primer, and 100 ng of template DNA. The *gyrB* cycle condition was an initial denaturation at 95 °C for 3 min, followed by 35 cycles of 94 °C for 30 s, 50 °C for 1 min and 72 °C for 1 min 30 s, and a final extension of 72 °C for 5 min. The amplification was performed in a T100 thermal cycler (Bio-Rad, Hercules, CA, USA). Direct sequencing of purified PCR products was performed by FirstBase (Malaysia).

#### 2.6.2. Water Quality

Table 3 summarises the water quality parameters of the farm during the experimental period. Most parameters were within the acceptable range based on Tookwinas [26] except the ammonia–nitrogen, which was high.

### 2.7. Statistical Analysis

The data were tabulated using Excel (Microsoft, Redmond, WA, USA). The normality and homogeneity of the variances were performed using Levene’s test. A two-tailed Student’s *t*-test with subsequent Bonferroni correction was used to determine the statistical significance of differences observed between the vaccinated and control groups using IBM SPSS Statistics 26 (SPSS 26.0 package, SPSS Inc., Chicago, IL, USA). The statistical significance was set at *p* < 0.05.

## 3. Results

### 3.1. Serum Systemic Antibody Response

The *Vibrio*-specific serum antibody (IgM) levels of the immunised fish were assessed by indirect ELISA from day 7 to day 70 post-primary vaccination (Figure 2, Figure 3 and Figure 4). Prior to vaccination, the antibody levels in serum samples of Asian seabass in both vaccinated and unvaccinated groups against *Vibrio harveyi*, *V. parahaemolyticus,* and *V. alginolyticus* were low (*p* > 0.05). Following oral immunisation with the *Vibrio harveyi* strain VH1 vaccine, the IgM levels from as early as day 7 of the vaccinated fish were significantly (*p* < 0.05) higher than the unvaccinated control group against all tested *Vibrio* spp. After subsequent administration of the first booster on day 14, the IgM levels of vaccinated groups against *V. harveyi* and *V. parahaemolyticus* increased significantly (*p* < 0.05) until day 35, when they reached a high value, while in *V. alginolyticus*, they increased until day 28, before slightly dropping until day 42. However, the IgM value was still significantly (*p* > 0.05) higher in the vaccinated group than in the unvaccinated control group. Following the second booster dose on day 42, the IgM levels against *V. harveyi* and *V. alginolyticus* increased significantly (*p* < 0.05) on day 49, while for *V. parahaemolyticus*, they increased on day 63. The antibody levels in the vaccinated group remained significantly higher (*p* < 0.05) than the unvaccinated control group at each time point in the vaccination period against *Vibrio harveyi*, *V. parahaemolyticus,* and *V. alginolyticus*. Antibody titres of the control group remained relatively stable and low throughout the experimental period.

### 3.2. Protection against Pathogenic Challenge

With respect to the protection level, all immunised Asian seabass groups exhibited varying degrees of protection against the pathogenic vibrio strains (*Vibrio harveyi, V. alginolyticus,* and *V. parahaemolyticus*; Figure 5). Asian seabass group (2) immunised with the oral *Vibrio harveyi* strain VH1 vaccine displayed a better survival rate than control non-immunised fish groups with an RPS of 70–85% (Table 4).

### 3.3. Field Study

#### 3.3.1. Weight Gain Effect and Feed Efficiency

Feed efficiency and weight gain were studied in large-scale field aquaculture of farm hybrid grouper, *Epinephelus fuscoguttus* × *Epinephelus lanceolatus* for four months (16 weeks or 112 days) with oral vaccination given at days 0, 14, and 42 for 5 consecutive days. The field study was initiated with the farm hybrid grouper 30.87 ± 3.65 g. After four months of monitoring, a stronger increase in body weight (248 ± 36.1 g) was measured for the *Vibrio harveyi* strain VH1 vaccine group. In contrast, the control group showed a bodyweight increase of 208 ± 21.5 g, indicating that the *Vibrio harveyi* strain VH1 feeding caused an approximately 22.56% weight gain (Figure 6). Total feed consumption was monitored during the 4-month experiment, and feed efficiencies were evaluated based on the relationships between feed consumption and weight gain. As shown in Table 5, the *Vibrio harveyi* strain VH1 vaccine group was 58.05% more efficient in feed conversion than that of the control, indicating that the vaccinated group required 58.05% less feed than the control group to produce a unit fish body weight.

#### 3.3.2. Rate of Survival

At the end of the 16-week study period, the survival rate of vaccinated fish was 79.75 ± 0.07%, significantly (*p* < 0.05) higher than the 65.1 ± 0.14% of control fish (Figure 7). Dead fish were found to suffer from severe scale drop, skin ulceration, and muscle necrosis (Figure 8). *Vibrio harveyi, V. alginolyticus, V. communis,* and *Photobacterium damselae* were recovered on TCBS agar and identified from the skin and kidney of the dead fish in both control and vaccinated groups.

## 4. Discussion

*Vibrio* species are Gram-negative bacteria responsible for vibriosis disease in marine fishes and are becoming a major threat to the aquaculture industry [27]. Vaccination, an alternative to antibiotics, has been proven to control infectious diseases more safely [28]. However, the progress of vaccine development against vibriosis has been slow due to the presence of different species of *Vibrios* that are environmentally and clinically important in aquatic environments and the antigenic diversities of the strains and serotypes [12,14]. The vaccines did not elicit protection against vibriosis infections caused by diverse strains.

A challenge trial was conducted 65 days post-primary vaccination in the current study. The RPS values obtained from the vaccinated group against *V. harveyi*, *V. alginolyticus*, and *V. parahaemolyticus* in Asian seabass, *Lates calcarifer* was higher (70–80%) compared with the control (0%). The *Vibrio*-specific antibody levels in the serum of vaccinated fish were significantly higher than in controls until day 70 post-primary vaccination, supporting the role of the vaccine in generating a protective response. It was expected that the vaccine administered orally would stimulate the immune response from various fronts in the fish, as does the pathogen, and that the antibody production in the cells would allow the stimulation of immune response in the fish, causing a strong response against the pathogen [29]. However, the bacterial challenge was carried out less than a month post final boost following previous studies dealing with oral vaccinations [30,31,32]. The timing of the challenge was too soon after vaccination to indicate long-term protection.

Moreover, since bacterial extracts have been shown to stimulate innate immunity in fish and potentially act as an immunostimulant, it may contribute to the rise of immunoglobulin levels. Giri et al. [33] reported that the rise in immunoglobulin levels is a short-term phenomenon attributable to immunostimulants. Due to the vaccine being at its early development and the current study only wanting to observe its early responses, further study needs to be conducted to determine whether the oral vaccine could provide longer-term protection to the fish.

Although Fraser et al. [34] concluded that vaccination would reduce the growth of fish due to an increased regular metabolic rate following continuous stimulation of the immune system, this current study found that feeding the farm hybrid grouper with the oral *V. harveyi* strain VH1 vaccine can improve the growth performance of the fish. Amar et al. [35] suggested that as fighting diseases and protecting against infections require a physiological cost, an ‘immune’ host could save energy for carrying and hosting pathogens, leaving more resources available for normal growth. Therefore, vaccination can promote growth by reducing the metabolic load of the immune response to infection.

A commercial adjuvant can be very expensive, especially for the commercial preparation of the vaccine. Thus, an alternative adjuvant that could provide good stimulation of immunity and subsequent protection at a cheaper price should be considered [36]. According to some studies, using cheap oilseed such as palm oil as an adjuvant for Newcastle disease virus (NDV) vaccination in chickens and caseous lymphadenitis vaccination in rats had successfully boosted immune protection while causing no negative effects [37,38]. Studies by Aminudin et al. [36] and Monir et al. [39] also observed high protection levels in orally immunised tilapia against *S. agalactiae* when palm oil was used as an adjuvant. Therefore, palm oil might be a possible adjuvant for fish vaccines, stimulating strong immunities at a lower cost, though its contribution to the vaccine response observed here was not studied.

In summary, the present study found that when the VH1 strain was inactivated and combined with the feed as an oral vaccine, it induced a specific antibody response and had a significant cross-reaction capacity against several pathogenic *Vibrios*. Furthermore, field application of the oral *V. harveyi* strain VH1 vaccine suggests that it can improve the growth performance and survival in farmed hybrid grouper, *Epinephelus fuscoguttus* × *Epinephelus lanceolatus.* This current vaccine was shown to potentially provide sufficient protection to the host, with a similar protective level to that of our previously developed polyvalent vaccine [20] and immersion vaccine [40] against vibriosis. Although further research is needed to examine the effect of the oral *V. harveyi* strain VH1 vaccine at the molecular level and in mucosal response, our results suggest that the oral *V. harveyi* strain VH1 vaccine is a potential broad, cross-protective vaccine candidate for vibriosis.

## Figures and Tables

**Figure 1 animals-12-00133-f001:**
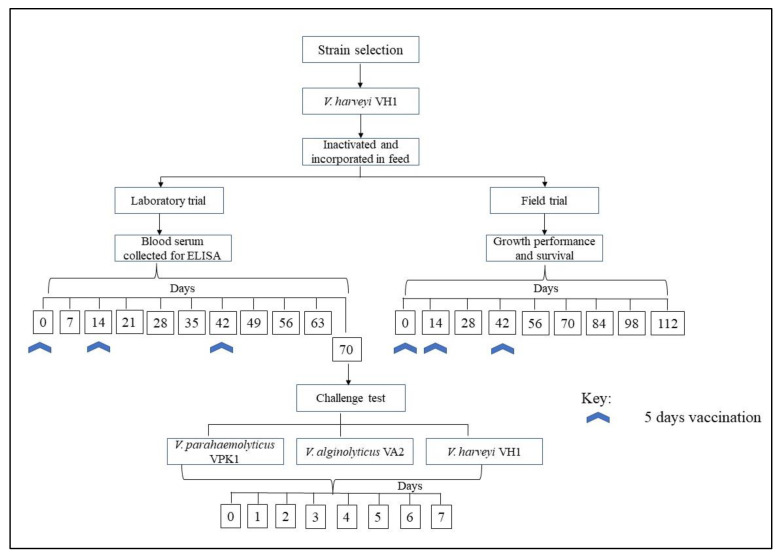
Overall graphical representation on strain selection, vaccine preparation and vaccination, and sampling schedule.

**Figure 2 animals-12-00133-f002:**
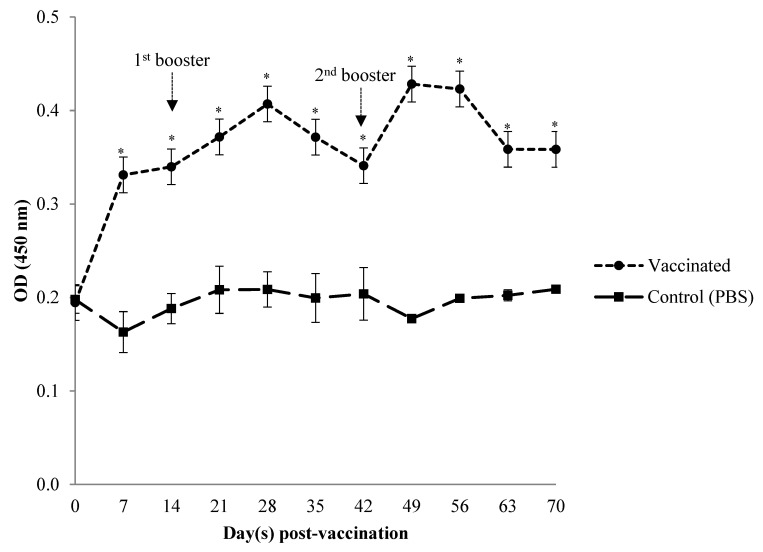
Antibody level of specific IgM in serum against *Vibrio harveyi* strain VH1 in Asian seabass, *Lates calcarifer* following oral vaccination with *Vibrio harveyi* strain VH1. Data are the mean ± SD from 6 fish per group at each time point. Asterisks stand for statistically significant differences (*p* < 0.05) between groups.

**Figure 3 animals-12-00133-f003:**
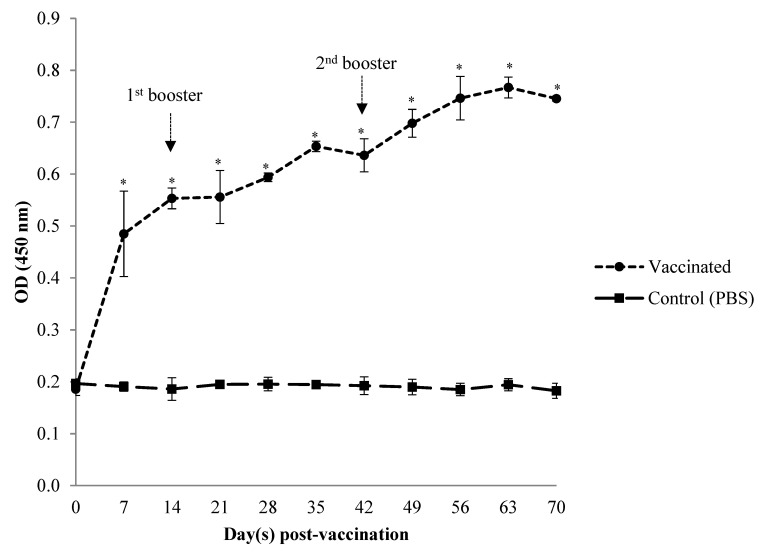
Antibody level of specific IgM in serum against *Vibrio parahaemolyticus* strain VPK1 in Asian seabass, *Lates calcarifer* following oral vaccination with *Vibrio harveyi* strain VH1. Data are the mean ± SD from 6 fish per group at each time point. Asterisks stand for statistically significant differences (*p* < 0.05) between groups.

**Figure 4 animals-12-00133-f004:**
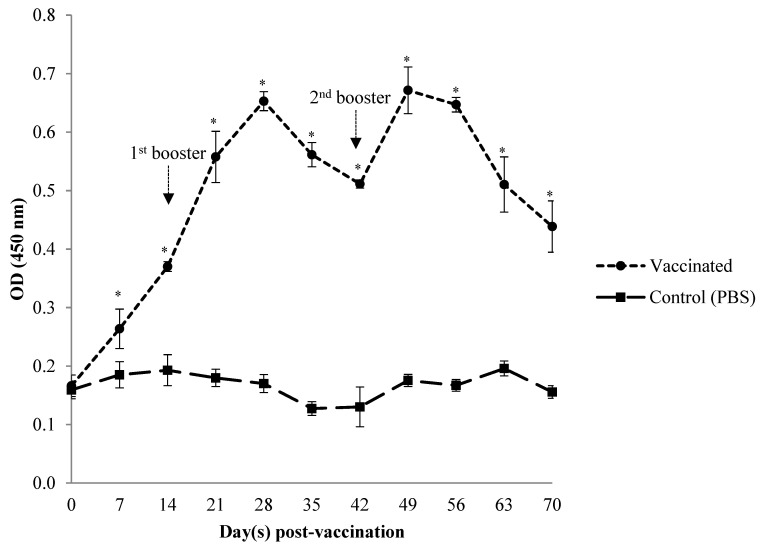
Antibody level of specific IgM in serum against *Vibrio alginolyticus* strain VA2 in Asian seabass, *Lates calcarifer* following oral vaccination with *Vibrio harveyi* strain VH1. Data are the mean ± SD from 6 fish per group at each time point. Asterisks stand for statistically significant differences (*p* < 0.05) between groups.

**Figure 5 animals-12-00133-f005:**
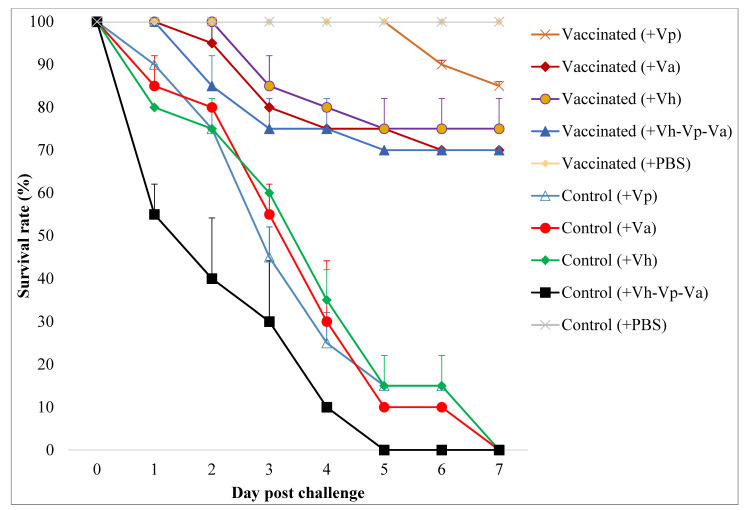
The survival rate of Asian seabass from vaccinated and unvaccinated control groups after challenging with PBS buffer (+PBS), single *V. parahaemolyticus* (+Vp), single *V. alginolyticus* (+Va), single *V. harveyi* (+Vh), or combined *V. harveyi*, *V. parahaemolyticus*, and *V. alginolyticus* (+Vh-Vp-Va). Each treatment was performed in duplicate with ten fish per challenge group (*n* = 10). The upper half part of the standard deviation bars is shown.

**Figure 6 animals-12-00133-f006:**
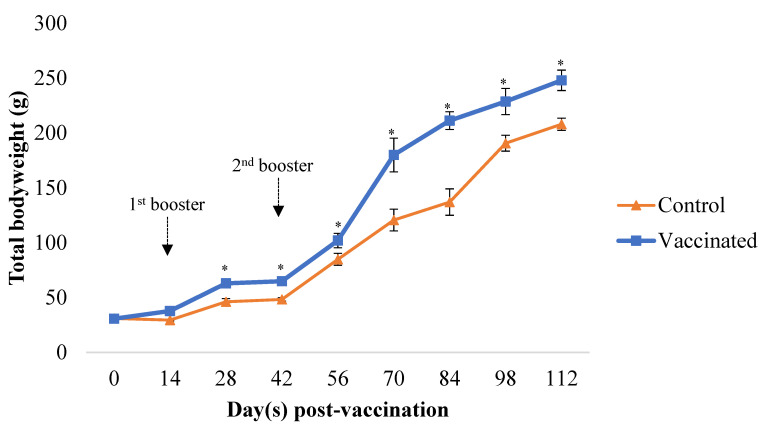
Farm hybrid grouper, *Epinephelus fuscoguttus* × *Epinephelus lanceolatus* body weight measured every two weeks for four months. Averaged values of the measurements were used to determine the mean bodyweight of farm hybrid grouper. Data are the mean ± SD from 15 fish per group at each time point. Asterisks stand for statistically significant differences (*p* < 0.05) between groups.

**Figure 7 animals-12-00133-f007:**
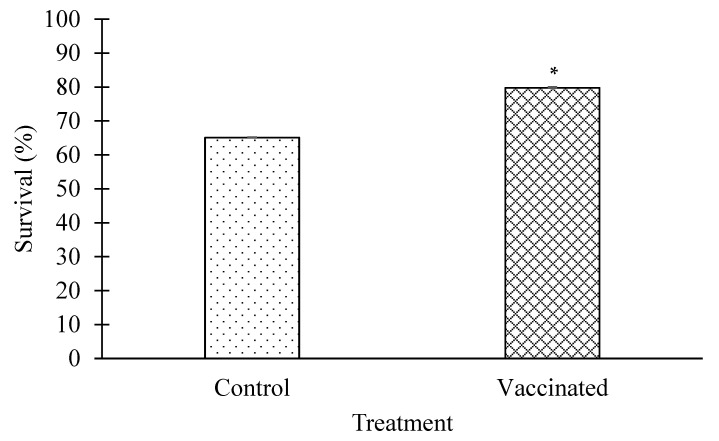
Percentage of survival between control and vaccinated group throughout the 112-day vaccination period. * Indicates a significant difference (*p < 0.05*) between the vaccinated and control groups.

**Figure 8 animals-12-00133-f008:**
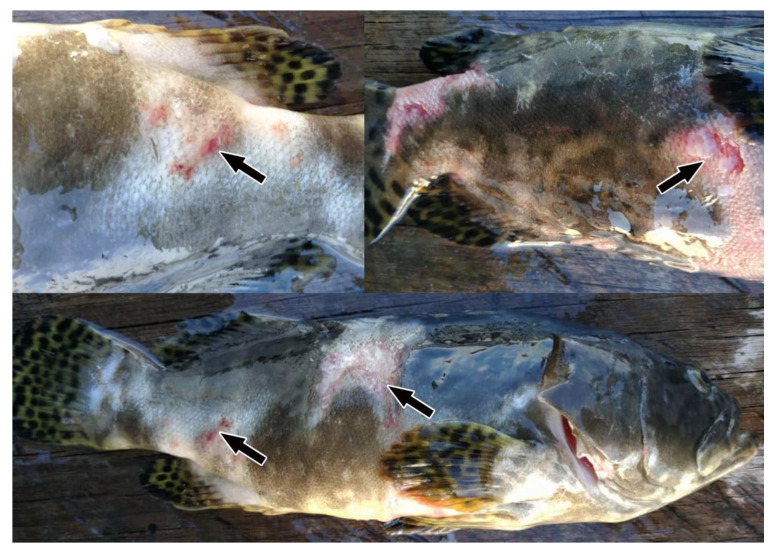
Severe scale loss, skin ulceration, and muscle necrosis were observed in naturally infected hybrid groupers with vibriosis. Arrows indicate the areas of infection on the fish body surface.

**Table 1 animals-12-00133-t001:** Bacterial strains genotype used in this study.

Bacterial Strain	Organ of Isolation	Host Species
*Vibrio harveyi* strain VH1	Skin, kidney, liver	*Epinephelus fuscoguttus*
*Vibrio alginolyticus* strain VA2	Liver, kidney	*Epinephelus fuscoguttus*
*Vibrio parahaemolyticus* strain VPK1	Liver	*Epinephelus fuscoguttus*

**Table 2 animals-12-00133-t002:** Primers used in PCR amplification of gyrB gene.

Primers	Primer Sequence (5′-3′)	Tm (°C)	Expected Size (bp)	Reference
gyrB_F	GAGAACCCGACAGAAGCGAAG	50.0	314	[25]
gyrB_R	CCTAGTGCGGTGATCAGTGTTG			

**Table 3 animals-12-00133-t003:** The mean ± SD and range of water quality parameters in cage farm area during the study period.

Parameter	Mean ± SD	* Range
Ammonia–nitrogen (mg/L)	0.03 ± 0.02	less than 0.02
pH (1–14)	8.00 ± 0.15	7.5–8.3
Temperature (°C)	30.45 ± 0.62	26–32
Salinity (ppt)	30.12 ± 1.33	10–31
Dissolved oxygen(mg/L)	4.71 ± 0.39	4.0–8.0

* Water quality parameter range was determined based on Tookwinas [26].

**Table 4 animals-12-00133-t004:** Comparative efficacy of oral *Vibrio harveyi* strain VH1 vaccine candidate against different *Vibrio* spp. in Asian seabass (*Lates calcarifer*).

Group	Bacterial Challenge	Number of Challenged Fish	Mortality (%)	RPS (%)
Control (unvaccinated)	PBS	20	0 ± 0.0	-
	*Vibrio harveyi* (Vh)	20	100 ± 0.0	-
	*V. alginolyticus* (Va)	20	100 ± 0.0	-
	*V. parahaemolyticus* (Vp)	20	100 ± 0.0	-
	Vh-Vp-Va	20	100 ± 0.0	-
Vaccinated	PBS	20	0 ± 0.0	-
	*Vibrio harveyi* (Vh)	20	25 ± 7.1	75
	*V. alginolyticus* (Va)	20	30 ± 0.0	70
	*V. parahaemolyticus* (Vp)	20	15 ± 7.1	85
	Vh-Vp-Va	20	30 ± 0.0	70

RPS, relative percent survival; -, not applicable. Each treatment was performed in duplicate with 10 fish per challenge group.

**Table 5 animals-12-00133-t005:** Hybrid grouper fed with *Vibrio harveyi* strain VH1 (1 × 10^7^ CFU/g feed) at 4% body weight according to the vaccination regime. Fifteen fish were randomly selected from each group every two weeks. *Vibrio harveyi* strain VH1 feeding group showed increased feed efficiencies.

Initial weight (g)	Control	31.00 ± 0.97
	Vaccinated	30.73 ± 0.92
Final weight (g)	Control	208 ± 5.54
	Vaccinated	248 ± 9.32
Average weight gain	Control	177.3 g/fish
	Vaccinated	217.3 g/fish
Survival (%)	Control	65.1 ± 0.14%
	Vaccinated	79.75 ± 0.07%
Total amount of feed given (kg)	Control	688.8 ± 13.22 kg
	Vaccinated	890.4 ± 18.99 kg
Feed efficiency	Control	7.39 ± 0.08 g feed/g growth
	Vaccinated	3.10 ± 0.09 g feed/g growth

## Data Availability

Not applicable.

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
