# Peer review of "Laboratory and Field Assessments of Oral Vibrio Vaccine Indicate the Potential for Protection against Vibriosis in Cultured Marine Fishes"

_animals, 2022, doi:10.3390/ani12020133_

Round 1

Reviewer 1 Report

This manuscript deals with the use of vaccines against Vibrio infections. The most interesting issue is the methodological approach, since the work consist of two stages: experimental and field. In that way, the results can be transferred quickly to the seafarms. It is a end-point work, from the lab to the company; I think this is the best way to assure the experimental results have positive effects on society and business.

Overall, the methods and results are well presented, including some images which help to understand better all the procedures. As the application was based on feeding, I think a more precise study on feed ingestion should be addressed (I have pointed out some comments in the attach). Nevertheless, it is clear that the vaccine application method was effective according to the final results.

I think the manuscript should be revised for minor errors (see attach) before publication.

Author Response

Dear reviewer, 

Thank you for your consideration. 

Reviewer 2 Report

The study shows some interesting results.  There are however a number of flaws, the most important are highlighted here and detailed comments are listed later under the different sections.

Unfortunately, the study continues to follow previous studies where challenge is conducted much too soon after last immunisation.  There is little evidence presented that the vaccine represents an effective vaccine providing long-term protection.  Here the controlled laboratory challenge is only 23 days – less than a month.  It is acknowledged that it is costly to maintain fish long-term in the aquarium, and that the initial early protection illustrated by the authors is important.  However, the study does not allow for evidence of long-term protection and this needs to be discussed in the manuscript.

There are issues with the presentation of the rabbit sera detection of Vibrio OMPs in that no preimmunization sera seems to have been collected and analysed from the different rabbits in terms of non-specific binding.

An ELIZA should be conducted using the vaccinated and control fish sera with the other Vibrio species against which heterologous antigenicity is claimed by the authors.

The data on the growth differences between vaccinated and control fish should be reanalysed – it does not seem correct to average all weights over time.

The field mortality over time in control and vaccinated fish should be presented.

INTRODUCTION

Lines 101-102: qualify the statement on protection by adding that they provide good protection under laboratory, i.p. administrated, challenge models.

The authors say that it is a newly developed oral vaccine candidate, but they have recently published a paper on the use of this vaccine in a multipathogen oral vaccine, as well as an i.p. delivered vaccine.  These studies should be mentioned.

MATERIALS AND METHODS

2.1. Bacterial strain selection

Lines 110-111: please add the volume of trypticase soya broth used to grow up bacteria.

2.1.4. Preparation of Antisera

There is no detail on why 8 rabbits were used – was this two rabbits per bacterial inoculum?  There is no detail on controls – e.g. injection of rabbits with Freund’s adjuvant only, or taking of blood samples from each individual rabbit prior to vaccination to account for non-specific binding and variation between rabbits.  This needs to be addressed to validate and better interpret the differences seen between Vibrio species in Western blot analyses.

Please give the concentration of Vibrio membrane protein loaded for each species on the SDS gel.

Lines 191-199: if this is a repeat of the methodology used to generated formalin inactivated bacteria for rabbit anti-sera production, then it could be shortened and the earlier method referred to, with any minor differences listed.

2.2. Preparation of feed vaccine

Please add the number of bacteria added per gram of feed and give the name of the commercial feed and the manufacturer.

Some detail on the feed quality analysis should be given or alternatively provide a reference detailing a similar analysis.

2.3. Fish vaccination and sample collection in laboratory trials

Additional information should be given on temperature during the trial, light regime, oxygen, the commercial name of the feed and supplier, or its composition if made inhouse.

Lines 219-220: the phrasing used in this sentence implies that vaccine was administrated for the full four weeks prior to challenge.  The text should be changed to reflect the experimental outline in figure 1.

In relation to the blood samples collected from the caudal vein, it should be clarified if these samples were taken non-lethally and the fish returned, or if the fish were killed.

Figure 1: the red colour on the figure does not represent to scale the 5 of 7 week days over which the vaccine was fed to fish. It would also be good to add to the figure arrows indicating when blood was collected, and also the week/date when the challenge experiment was terminated.

I also think that it might be clearer to indicate days post start of experiment/post vaccination rather than weeks e.g. in figure 1 a booster vaccination at 6 weeks and challenge at 10 weeks could be interpreted as a four week interval whereas in reality it is closer to 3 weeks between vaccination and challenge.

2.4. Determination of Vibrio harveyi strain VH1-specific serum antibody production

Line 237: mucus and gut lavage – there is no mention of collection of these samples, and data is not presented.  Please provide the data.

What is the reason for conducting the ELISA at 37°C?

2.5. Experimental Challenge of Vibrio sp.

Please give in the text the number of fish challenged.

Was the weight of the fish recorded in the experimental challenge?  If so then it could be provided– to compare with observations in the field trial.

2.6. Field trial

Line 265: why only PBS in the control fish versus PBS+palm oil in the lab experimental controls, moreover the legend in Figure 2 says control fish feed contained PBS+palm oil?

There is no mention of Vibrio re-isolation from fish mortalities – though this is mentioned in results.  Please add the information on the sampling e.g tissues – and detection/enumeration method.

2.7 Statistical analysis: please provide some additional information on the statistical analyses - e.g. did the analysis need to be corrected and if so what correction method was used?

RESULTS

3.1. Strain selection

Line 301: just to check - is it correct to refer to P. damselae as a Vibrio species?

Lines 301-302: it is difficult to understand what is meant by the last part of this sentence, please rephrase.

Figure 3: Even though the bacterial species represented in each lane is provided in the text it would be helpful to also place in the figure legend.  Information should also be provided on what is represented by the different coloured arrows.

Figure 4: why does it seem that extracts from different gels were isolated and placed together? It would be more convincing if the results of the single gel containing the four bacterial species and exposed to a single anti-serum was presented.  The labelling in the figure seems to have moved – i.e. b,d labels.  It might be better to have different symbols instead of asterisks with different colours – in case readers print article in black and white.

There are no controls presented for staining with rabbit serum taken prior to immunisation.  This is important to present as otherwise the results are questionable.

3.2. Serum systemic antibody response

Line 327: please clarify in the text that this is Vibrio specific IgM.

Line 328: 16 weeks is indicated instead of 10 weeks.

Figure 5: some of the letters indicating significance need to be realigned.

3.2. Protection against pathogenic challenge.

Figure 6: the vaccinated+PBS data seems to be missing from the graph. The legend says that only 10 fish per group remained for the challenge.  It is not clear on how this number was arrived at based on the starting numbers given in the materials and methods.

3.3.1. Weight gain effect and feed efficiency

Lines 362-363: the text in these lines reads as if the vaccine was administrated over the full four months.  It needs to be modified.

Figure 8: the text implies that all measurements over time were combined.  It may be more correct to plot the average and variation for each time point, and to statistically compare controls and vaccinated for each time point, or alternatively, to compare for the last time point only. 

Additionally the results do not look significant – please check the statistical analysis again.

Similarly in table 3, the average weight gain should be for the final time point.  It would be interesting to plot/compare the feed conversion rate per week – e.g. is the weight gain/conversion rate stable over time, or is it higher in vaccinated fish for just a short period after vaccination/boost and then reduces again?

3.3.3. Rate of survival

Line 384: please clarify in the text that Vibrio species were isolated from dead fish in both the control and vaccinated groups. 

The data on field mortality in the vaccinated and control fish should also be plotted over time to see how mortality levels compared in the two groups over time – i.e. indication of whether the vaccine response remain effective as time since vaccination increased.

Table 4: perhaps in supplementary data the plots of data for the different parameters over time could be given.  Since temperature (or other environmental/water parameters) at vaccination/just after vaccination may affect efficacy, it might be useful to help interpret results in field data generated by future studies.

DISCUSSION

Line 427: it should also be mentioned that the challenge was only 3 weeks post final boost.  The issue of the timing of the challenge so soon after vaccination needs to be addressed in that it cannot indicate long term protection and could even represent innate immunostimulation.  This needs to be discussed in terms of what other authors have found for oral bacterial vaccines.

Line 425-426: to confirm cross reactivity further, ELISA should be conducted with the other bacterial species using fish serum from the lab challenge.  This should be done to confirm choice of V. harveyi for heterologous antigenicity.

Lines 433-436: however mucosal response was not examined - this should be reiterated in the text in case the work is quoted as supporting development of a mucosal response.

Lines 459-462: if the field control were administrated palm oil in their feed, as indicated in the legend of figure 2, then is this a logical reason for the increase in weight in vaccinated fish only?

The use of palm oil needs to be discussed further.  There is no information presented on whether it is commonly/previously used as an adjuvant and whether there has been analysis of any side effects on the fish species in question.  Are there any reports of it being an immunostimulant on its own?

The authors have published on this strain of V. harveyi previously in i.p. and combined oral vaccines.  Some discussion on the comparative efficacies observed in relation to protection should be provided.  They should also make some comment on the existence of any other studies on V. harveyi oral vaccines for fish.

Author Response

Cover letter

On behalf of all the authors I, Aslah Mohamad states that there is no conflict of interest about the study submitted to the journal for possible publication.

Many thanks

Sequel to your mail to us on the review of our manuscript title “Laboratory and Field Assessments of oral Vibrio vaccine provides good protection against vibriosis in cultured marine fishes” the following correction and rebuttals were made:

Comments and suggestions from the reviewer (Reviewer 2)

General comments

The study shows some interesting results.  There are however a number of flaws, the most important are highlighted here and detailed comments are listed later under the different sections.

Unfortunately, the study continues to follow previous studies where challenge is conducted much too soon after last immunisation.  There is little evidence presented that the vaccine represents an effective vaccine providing long-term protection.  Here the controlled laboratory challenge is only 23 days – less than a month.  It is acknowledged that it is costly to maintain fish long-term in the aquarium, and that the initial early protection illustrated by the authors is important.  However, the study does not allow for evidence of long-term protection and this needs to be discussed in the manuscript.

  • Thank you for your comments. As the vaccine is at its early development, we are still observing its early response and protection against bacterial infection. However, we had already revised and included the issues regarding the long-term protection in the discussion part.

There are issues with the presentation of the rabbit sera detection of Vibrio OMPs in that no preimmunization sera seems to have been collected and analysed from the different rabbits in terms of non-specific binding.

  • Thank you again for the comment. Preimmune sera was collected and analysed from different rabbits but not being mentioned in the manuscript. Thus, we have added the statement for the preimmune sera collection in Line 162. However, no band was detected by the preimmune sera.

An ELIZA should be conducted using the vaccinated and control fish sera with the other Vibrio species against which heterologous antigenicity is claimed by the authors.

  • We do have the results of antibody level against other Vibrio species which we aimed to publish in a different manuscript focusing systemic and mucosal immune response post-vaccination with the oral feed. However, we have included ELISA results against parahaemolyticus and V. alginolyticus in this manuscript following reviewer’s comment.

The data on the growth differences between vaccinated and control fish should be reanalysed – it does not seem correct to average all weights over time.

  • The data on the growth differences was revised.

The field mortality over time in control and vaccinated fish should be presented.

  • The field mortality over time in different group is presented in Line xx.

INTRODUCTION

Lines 101-102: qualify the statement on protection by adding that they provide good protection under laboratory, i.p. administrated, challenge models.

  • The statement was added.

The authors say that it is a newly developed oral vaccine candidate, but they have recently published a paper on the use of this vaccine in a multipathogen oral vaccine, as well as an i.p. delivered vaccine.  These studies should be mentioned.

  • The statement newly developed was removed.

MATERIALS AND METHODS

2.1. Bacterial strain selection

Lines 110-111: please add the volume of trypticase soya broth used to grow up bacteria.

  • 250 mL of trypticase soya broth was used to grow up bacteria and mentioned in Line 111. Thank you for noticing the missing volume.

2.1.4. Preparation of Antisera

There is no detail on why 8 rabbits were used – was this two rabbits per bacterial inoculum?  There is no detail on controls – e.g. injection of rabbits with Freund’s adjuvant only, or taking of blood samples from each individual rabbit prior to vaccination to account for non-specific binding and variation between rabbits.  This needs to be addressed to validate and better interpret the differences seen between Vibrio species in Western blot analyses.

  • Only 8 rabbits were used for the antisera preparation. Each two of the rabbits were immunized with one ml of the formalin- killed whole cells of each Vibrio Preimmune sera was collected and analysed from the rabbits and used as control prior to vaccination to account for non-specific binding and variation between rabbits. Collection of preimmune sera as a control for non-specific binding was mentioned in Line 162.

Please give the concentration of Vibrio membrane protein loaded for each species on the SDS gel.

  • The concentration was put in Line 168

Lines 191-199: if this is a repeat of the methodology used to generated formalin inactivated bacteria for rabbit anti-sera production, then it could be shortened and the earlier method referred to, with any minor differences listed.

  • The methodology was revised following recommendations.

2.2. Preparation of feed vaccine

Please add the number of bacteria added per gram of feed and give the name of the commercial feed and the manufacturer.

  • The method was revised according to reviewer’s suggestion in Lines 207-215.

Some detail on the feed quality analysis should be given or alternatively provide a reference detailing a similar analysis.

  • Details on feed quality analysis was given according to reviewer’s suggestion in Lines 216-217.

2.3. Fish vaccination and sample collection in laboratory trials

Additional information should be given on temperature during the trial, light regime, oxygen, the commercial name of the feed and supplier, or its composition if made inhouse.

  • The method was revised according to reviewer’s suggestion in Lines 232-235

Lines 219-220: the phrasing used in this sentence implies that vaccine was administrated for the full four weeks prior to challenge.  The text should be changed to reflect the experimental outline in figure 1.

  • The method was revised accordingly.

In relation to the blood samples collected from the caudal vein, it should be clarified if these samples were taken non-lethally and the fish returned, or if the fish were killed.

  • The method was revised accordingly.

Figure 1: the red colour on the figure does not represent to scale the 5 of 7 week days over which the vaccine was fed to fish. It would also be good to add to the figure arrows indicating when blood was collected, and also the week/date when the challenge experiment was terminated.

  • The figure was revised according to reviewer’s suggestion.

I also think that it might be clearer to indicate days post start of experiment/post vaccination rather than weeks e.g. in figure 1 a booster vaccination at 6 weeks and challenge at 10 weeks could be interpreted as a four week interval whereas in reality it is closer to 3 weeks between vaccination and challenge.

  • Week was substitute with day

2.4. Determination of Vibrio harveyi strain VH1-specific serum antibody production

Line 237: mucus and gut lavage – there is no mention of collection of these samples, and data is not presented.  Please provide the data.

  • The text mucus and gut lavage was removed as it does not mention elsewhere for mucus and gut lavage collection.

What is the reason for conducting the ELISA at 37°C?

  • The ELISA procedure was conducted at 37°C due to the temperature provide more stable absorbance reading than the suggested temperature (22°C). Not only that, this temperature also provides shorter incubation period than the lower temperature.

2.5. Experimental Challenge of Vibrio sp.

Please give in the text the number of fish challenged.

  • The number of challenged fish was mentioned in Line 267

Was the weight of the fish recorded in the experimental challenge?  If so then it could be provided– to compare with observations in the field trial.

  • Yes, we agree. However, no weight data was collected during the experimental challenge.

2.6. Field trial

Line 265: why only PBS in the control fish versus PBS+palm oil in the lab experimental controls, moreover the legend in Figure 2 says control fish feed contained PBS+palm oil?

  • The method was revised accordingly. Control feed contained PBS and palm oil

There is no mention of Vibrio re-isolation from fish mortalities – though this is mentioned in results.  Please add the information on the sampling e.g tissues – and detection/enumeration method.

  • The method was revised accordingly.

2.7 Statistical analysis: please provide some additional information on the statistical analyses - e.g. did the analysis need to be corrected and if so what correction method was used?

  • The statistical analysis was revised accordingly.

RESULTS

3.1. Strain selection

Line 301: just to check - is it correct to refer to P. damselae as a Vibrio species?

  • Photobacterium damselae was previously known as Vibrio damsela is part of Vibrionaceae family which have been associated with health problems of marine animals similar to Vibrio parahaemolyticus, alginolyticus, V. harveyi, V. owensii, and V. campbellii (Ina-Salwany et al., 2019).

Lines 301-302: it is difficult to understand what is meant by the last part of this sentence, please rephrase.

  • The last part in Lines 301-302 was rephrased.

Figure 3: Even though the bacterial species represented in each lane is provided in the text it would be helpful to also place in the figure legend.  Information should also be provided on what is represented by the different coloured arrows.

  • Information was provided in the figure legend.

Figure 4: why does it seem that extracts from different gels were isolated and placed together? It would be more convincing if the results of the single gel containing the four bacterial species and exposed to a single anti-serum was presented.  The labelling in the figure seems to have moved – i.e. b,d labels.  It might be better to have different symbols instead of asterisks with different colours – in case readers print article in black and white.

  • The gels were run separately between different bacterial species. The figure was improved according to reviewer’s suggestion.

There are no controls presented for staining with rabbit serum taken prior to immunisation.  This is important to present as otherwise the results are questionable.

  • Pre-immune sera collection and band visibility were mentioned in the text.

3.2. Serum systemic antibody response

Line 327: please clarify in the text that this is Vibrio specific IgM.

  • The text was clarified accordingly.

Line 328: 16 weeks is indicated instead of 10 weeks.

  • It should be 10 weeks. Thank you for noticing this.

Figure 5: some of the letters indicating significance need to be realigned.

  • The letter indicating significance was realigned.

3.2. Protection against pathogenic challenge.

Figure 6: the vaccinated+PBS data seems to be missing from the graph. The legend says that only 10 fish per group remained for the challenge.  It is not clear on how this number was arrived at based on the starting numbers given in the materials and methods.

  • The figure and legend was revised accordingly

3.3.1. Weight gain effect and feed efficiency

Lines 362-363: the text in these lines reads as if the vaccine was administrated over the full four months.  It needs to be modified.

  • The text was revised accordingly.

Figure 8: the text implies that all measurements over time were combined.  It may be more correct to plot the average and variation for each time point, and to statistically compare controls and vaccinated for each time point, or alternatively, to compare for the last time point only. 

  • The figure was revised and changed based on each time point.

Additionally the results do not look significant – please check the statistical analysis again.

  • The results was re-run based on time-point following reviewer’s recommendations.

Similarly in table 3, the average weight gain should be for the final time point.  It would be interesting to plot/compare the feed conversion rate per week – e.g. is the weight gain/conversion rate stable over time, or is it higher in vaccinated fish for just a short period after vaccination/boost and then reduces again?

  • The results was revised accordingly.

3.3.3. Rate of survival

Line 384: please clarify in the text that Vibrio species were isolated from dead fish in both the control and vaccinated groups. 

  • The text mentioned in Line 384 was revised and clarified in Line xx.

The data on field mortality in the vaccinated and control fish should also be plotted over time to see how mortality levels compared in the two groups over time – i.e. indication of whether the vaccine response remain effective as time since vaccination increased.

  • Data on field mortality based on time-point was recorded based on visibly daily mortality. Actual number of survived fish calculated at the end of the vaccination period.

Table 4: perhaps in supplementary data the plots of data for the different parameters over time could be given.  Since temperature (or other environmental/water parameters) at vaccination/just after vaccination may affect efficacy, it might be useful to help interpret results in field data generated by future studies.

  • Data on different parameters over time is included in the supplementary data.

DISCUSSION

Line 427: it should also be mentioned that the challenge was only 3 weeks post final boost.  The issue of the timing of the challenge so soon after vaccination needs to be addressed in that it cannot indicate long term protection and could even represent innate immunostimulation.  This needs to be discussed in terms of what other authors have found for oral bacterial vaccines.

  • The discussion was revised as suggested by reviewer.

Line 425-426: to confirm cross reactivity further, ELISA should be conducted with the other bacterial species using fish serum from the lab challenge.  This should be done to confirm choice of V. harveyi for heterologous antigenicity.

  • ELISA was conducted with other bacterial species. The discussion was revised as suggested by reviewer

Lines 433-436: however mucosal response was not examined - this should be reiterated in the text in case the work is quoted as supporting development of a mucosal response.

  • The discussion was revised as suggested by reviewer.

Lines 459-462: if the field control were administrated palm oil in their feed, as indicated in the legend of figure 2, then is this a logical reason for the increase in weight in vaccinated fish only?

  • The discussion was revised.

The use of palm oil needs to be discussed further.  There is no information presented on whether it is commonly/previously used as an adjuvant and whether there has been analysis of any side effects on the fish species in question.  Are there any reports of it being an immunostimulant on its own?

  • The use of palm oil was further discussed.

The authors have published on this strain of V. harveyi previously in i.p. and combined oral vaccines.  Some discussion on the comparative efficacies observed in relation to protection should be provided.  They should also make some comment on the existence of any other studies on V. harveyi oral vaccines for fish.

  • The vaccine in the current study was compared with the vaccine from our previous studies against vibriosis.

Reviewer 3 Report

The paper of Mohamad et al. is centered around a very interesting and promising topic for all those related to the aquaculture industry and research, that is efficacious vaccine development. Many reviews have been published in the last 2-3 years depicting the state of the art of commercial and experimental vaccine research, highlighting the interest surrounding the subject.

The paper is potentially interesting as it also included a field-experiment in addition to the usual small-scale lab experiments that are not necessarily reflective of real-world scenarios.

I believe the paper could benefit greatly from the comments below. Some are purely stylistic, other point out to methodological drawbacks that authors need to clarify. English grammar and style must be slightly revised throughout the manuscript.

All line numbers refer to the pdf version of the submission.

L 130: Vibrio strains

L131: ultracentrifuged

L158: “Each of the rabbits” – there were four whole-cell protein preparations, so it must be two rabbits/Vibrio strain

L170: “All polyclonal rabbit antisera were incubated”. Rephrase, the membrane was incubated with…

L170-176 are confusing. It appears that the membrane was incubated twice

L188: “In this study, we used a specific strain”. It’s not clear why have you used this strain for. If your experimental method of the present paper is based on past observation please cite the reference

L196: “Bacterial cells of each strain were then inactivated”… Earlier you wrote that only strain VH1 was used for vaccine production, while here the sentence implies that all strain were grown and inactivated.

L204: It’s the first time I personally see palm oil being used as adjuvant. Authors should justify this choice and provide adequate reference for it.

L209: provide more details on the patent or at least the patent agency to which it was filed.

Fig1 is actually not very insightful. It might be improved by providing an overall graphical representation of the study rationale, with a summary of vaccinations, challenges and analysis performed on lab and field specimens per time point. Readers could benefit from a clearer view on the experimental procedure and source of bacterial material used (whole cell extract, OMP, FKC Vibrio strains)

L212: University

L213: Provide some sort of variability measure of the average weight

L217: Control feed contained PBS and palm oil, not just PBS – you wrote this at line 206

L218: “Single and double boosters were given…”. How could these two vaccination strategies (one with a single booster, a second with two boosters) be tested if only one vaccination group (in duplicates) was used? Did each replicate received a single booster and a double booster?

L254: How can you safely declare that no abnormal behaviour was associated to the challenge procedure if fish were anaesthetized in that moment?

L282: Were t test assumptions verified for all three datasets?

Fig3: I believe that Figure 3 doesn’t meet quality standards. Also, if I’m not wrong, SDS-PAGE or Western blot images are requested in their non-cropped original versions by all MDPI journals. Also, not all bands were reported in the text when describing obtained results or pointed by arrows. Why?

Fig4: Same comment as above plus single images are misplaced in the panel and the meaning of black and red asterisks are not evincible from the legend. Also, figure legend should read “Immunoblot profiles of outer membrane proteins (OMPs) from four Vibrio spp and P. damselae….”

Fig. 5: The first booster at week 2 and the second booster at week 6 aren’t properly indicated in the plot. Correspondence between IgM values/trends in the text sometime are not reflected in the plot as well.

L345: With respect and Asian

Table 2: 10 fish from vaccinated/non vaccinated groups were challenged in duplicates. This means 20 fish per group per challenge type. Colum 3 should therefore be amended. Values reported in column 4 don’t make any biological sense (how could 2.5 fish be dead?). I assume this is an average±variability (SD or SE). Not too sure about the values in the last two columns. If 20 and not 10 fish were challenged per group per challenge type, mortality cannot just be number of dead fish by 10.

L362: Duration of field trial has always been expressed in weeks so far, please be consistent. Also, as is, it seems that fish were fed VH1-containing feed for the whole 4 weeks, which according to Fig. 2 is not the case.

L366: I don’t get the possible relation of control/vaccinated feed and weight gain. Do authors attribute the weight gain exclusively to the vaccinated feed? Why is there such a discrepancy in the total amount of feed given to the two experimental groups? Were fish randomly sampled and weighed during the 16-week experiment to re-calibrate feed amounts to administer based on the 4% BW

Fig.8: Please show all data points per each boxplot and report all details of the test statistics, not only the approximate p value. Also, the standard box plot visualization consists in the 25-75 quartile, aka Tukey in some statistical software, not min to max.

Line 381: Did you employ percentage data directly in the t test analysis?

Fig.9: Normally asterisks are overlaid on the bar that represents the statistically significantly different group compared to the control.

L388-389: Rephrase for clarity

L413: It’s not clear to me why authors group keep on grouping P. damselae under the Vibrio spp. umbrella of species.

L435: In an attempt to justify ELISA results, the author state they expected systemic and mucosal IgM response to be stimulated. However, structural differences exist between mucosal and systemic IgM (polymer vs monomer) and no detail was given as to the ability of the Aquatic Diagnostics IgM monoclonal antibody employed to discriminate between the two. Systemic and mucosal IgM responses cannot be concluded.

L460: Again an uncertainty regarding weight increase. Authors justify that palm oil was responsible for weight increase. However, the control group received a feed supplemented with PBS AND palm oil as well. Their conclusion is not backed up by experimental design or results. Also, an excess of fatty acids may result in undesirable conditions i.e. dyslipidemia. Rather, were animal sacrificed at the end of the field survey and checked for any internal condition of the liver? Was the liver swollen?

Thank you and best regards

Author Response

Cover letter

On behalf of all the authors I, Aslah Mohamad states that there is no conflict of interest about the study submitted to the journal for possible publication.

Many thanks

Sequel to your mail to us on the review of our manuscript title “Laboratory and Field Assessments of oral Vibrio vaccine provides good protection against vibriosis in cultured marine fishes” the following correction and rebuttals were made:

Comments and suggestions from the reviewer (Reviewer 3)

All line numbers refer to the pdf version of the submission.

L 130: Vibrio strains

  • The word Vibrio was italic

L131: ultracentrifuged

  • The word was changed

L158: “Each of the rabbits” – there were four whole-cell protein preparations, so it must be two rabbits/Vibrio strain

  • Yes, true. The sentences were revised accordingly.

L170: “All polyclonal rabbit antisera were incubated”. Rephrase, the membrane was incubated with…

  • The method was revised accordingly

L170-176 are confusing. It appears that the membrane was incubated twice

  • The method was revised accordingly

L188: “In this study, we used a specific strain”. It’s not clear why have you used this strain for. If your experimental method of the present paper is based on past observation please cite the reference

  • The method was revised accordingly

L196: “Bacterial cells of each strain were then inactivated”… Earlier you wrote that only strain VH1 was used for vaccine production, while here the sentence implies that all strain were grown and inactivated.

  • The method was revised accordingly

L204: It’s the first time I personally see palm oil being used as adjuvant. Authors should justify this choice and provide adequate reference for it.

  • The use of palm oil was further discussed in discussion part.

L209: provide more details on the patent or at least the patent agency to which it was filed.

  • The patent agency (MyIPO, Malaysia) was included in the text.

Fig1 is actually not very insightful. It might be improved by providing an overall graphical representation of the study rationale, with a summary of vaccinations, challenges and analysis performed on lab and field specimens per time point. Readers could benefit from a clearer view on the experimental procedure and source of bacterial material used (whole cell extract, OMP, FKC Vibrio strains)

  • Figure 1 was revised accordingly.

L212: University

  • The word Universiti is remain because it is a noun.

L213: Provide some sort of variability measure of the average weight

  • Variability measure was included in the text.

L217: Control feed contained PBS and palm oil, not just PBS – you wrote this at line 206

- Yes, thank you for noticing this. The word “palm oil” was included.  

L218: “Single and double boosters were given…”. How could these two vaccination strategies (one with a single booster, a second with two boosters) be tested if only one vaccination group (in duplicates) was used? Did each replicate received a single booster and a double booster?

  • The vaccinated group received two booster doses (first and second).

L254: How can you safely declare that no abnormal behaviour was associated to the challenge procedure if fish were anaesthetized in that moment?

  • The method was revised accordingly.

L282: Were t test assumptions verified for all three datasets?

  • T-test assumptions was used for all datasets.

Fig3: I believe that Figure 3 doesn’t meet quality standards. Also, if I’m not wrong, SDS-PAGE or Western blot images are requested in their non-cropped original versions by all MDPI journals. Also, not all bands were reported in the text when describing obtained results or pointed by arrows. Why?

  • The figure was revised accordingly. We only have that figure at the moment.

Fig4: Same comment as above plus single images are misplaced in the panel and the meaning of black and red asterisks are not evincible from the legend. Also, figure legend should read “Immunoblot profiles of outer membrane proteins (OMPs) from four Vibrio spp and P. damselae….”

  • The figure was revised accordingly.

Fig. 5: The first booster at week 2 and the second booster at week 6 aren’t properly indicated in the plot. Correspondence between IgM values/trends in the text sometime are not reflected in the plot as well.

  • The figure was revised accordingly.

L345: With respect and Asian

  • The typo was corrected

Table 2: 10 fish from vaccinated/non vaccinated groups were challenged in duplicates. This means 20 fish per group per challenge type. Colum 3 should therefore be amended. Values reported in column 4 don’t make any biological sense (how could 2.5 fish be dead?). I assume this is an average±variability (SD or SE). Not too sure about the values in the last two columns. If 20 and not 10 fish were challenged per group per challenge type, mortality cannot just be number of dead fish by 10.

  • Table 2 was revised accordingly

L362: Duration of field trial has always been expressed in weeks so far, please be consistent. Also, as is, it seems that fish were fed VH1-containing feed for the whole 4 weeks, which according to Fig. 2 is not the case.

  • Duration of vaccination trial was fixed as day

L366: I don’t get the possible relation of control/vaccinated feed and weight gain. Do authors attribute the weight gain exclusively to the vaccinated feed? Why is there such a discrepancy in the total amount of feed given to the two experimental groups? Were fish randomly sampled and weighed during the 16-week experiment to re-calibrate feed amounts to administer based on the 4% BW

  • The feed given was based on the weight and survival of the fish in every 14-days sampling.

Fig.8: Please show all data points per each boxplot and report all details of the test statistics, not only the approximate p value. Also, the standard box plot visualization consists in the 25-75 quartile, aka Tukey in some statistical software, not min to max.

  • The figure was changed to weight performance in every 14-days sampling.

Line 381: Did you employ percentage data directly in the t test analysis?

  • The data emplyed in the t test is based on the number of survived fish.

Fig.9: Normally asterisks are overlaid on the bar that represents the statistically significantly different group compared to the control.

  • The figure was revised accordingly

L388-389: Rephrase for clarity

  • The line was rephrased accordingly.

L413: It’s not clear to me why authors group keep on grouping P. damselae under the Vibrio spp. umbrella of species.

  • The line was revised accordingly. However Photobacterium damselae was previously known as Vibrio damsela and part of Vibrionaceae family which have been associated with health problems of marine animals similar to Vibrio parahaemolyticus, alginolyticus, V. harveyi, V. owensii, and V. campbellii (Ina-Salwany et al., 2019).

L435: In an attempt to justify ELISA results, the author state they expected systemic and mucosal IgM response to be stimulated. However, structural differences exist between mucosal and systemic IgM (polymer vs monomer) and no detail was given as to the ability of the Aquatic Diagnostics IgM monoclonal antibody employed to discriminate between the two. Systemic and mucosal IgM responses cannot be concluded.

  • The discussion was revised accordingly

L460: Again an uncertainty regarding weight increase. Authors justify that palm oil was responsible for weight increase. However, the control group received a feed supplemented with PBS AND palm oil as well. Their conclusion is not backed up by experimental design or results. Also, an excess of fatty acids may result in undesirable conditions i.e. dyslipidemia. Rather, were animal sacrificed at the end of the field survey and checked for any internal condition of the liver? Was the liver swollen?

  • The discussion was revised accordingly

Round 2

Reviewer 2 Report

The manuscript has been much improved by the authors with additional details provided on methodology and discussion. The additional analysis of weight at each time point and the addition of data from VH1 vaccinated fish on Vibrio specific antibody levels against the other Vibrio species is much appreciated and adds interesting information to the paper.

There are a number of relatively minor comments to address in the attached manuscript word document, and some suggested edits to the language.

Unfortunately, however, there remain two important issues to address.

One is the statistical analysis where multiple comparisons have been made, potentially introducing type I errors, and it is not clear if the analysis has taken this into account by including a correction factor. The authors should clearly show that their analysis deals with this, and indicate any approach used for correction.

As indicated in the previous review, additional work is required regarding the western blot results demonstrating cross antigenicity of the different anti-sera.  The authors need to show results for the pre-immune sera controls.  They have mentioned that these controls were run, but the results need to be presented.   

Moreover, they need to test the antisera against Freund’s complete adjuvant.  This adjuvant contains Mycobacterium and it may be possible that some of the cross reactivity is due to this factor rather than the Vibrio harveyi VH1, moreover as the 32 kDa band highlighted is detected by all sera.  The cross reactivity of the VH1 vaccine is one of the important attributes of the selected vaccine and an important part of the paper.  Therefore, this needs to be tested again with pre-immunisation and adjuvant-only controls, and alongside again the Vibrio extracts to act as positive controls for the original results – and the results need to be shown.

If this can be done then the paper is acceptable for publication, but without the above additional work, then the conclusions on the VH1 cross reactivity are not robust.

Author Response

Cover letter

On behalf of all the authors I, Aslah Mohamad states that there is no conflict of interest about the study submitted to the journal for possible publication.

Many thanks

Sequel to your mail to us on the review of our manuscript title “Laboratory and Field Assessments of oral Vibrio vaccine provides good protection against vibriosis in cultured marine fishes” the following correction and rebuttals were made:

Comments and suggestions from the reviewer (Reviewer 2-Second revision)

General comments

  1. One is the statistical analysis where multiple comparisons have been made, potentially introducing type I errors, and it is not clear if the analysis has taken this into account by including a correction factor. The authors should clearly show that their analysis deals with this, and indicate any approach used for correction.

-Type 1 errors were avoided with Bonferroni corrections (P=0.006).

  1. As indicated in the previous review, additional work is required regarding the western blot results demonstrating cross antigenicity of the different anti-sera.  The authors need to show results for the pre-immune sera controls.  They have mentioned that these controls were run, but the results need to be presented.

-The results were included in the figure

https://drive.google.com/file/d/16WgDurv07ERl6vjFg3CwdF2ZZlyIz14v/view?usp=sharing

  1. Moreover, they need to test the antisera against Freund’s complete adjuvant.  This adjuvant contains Mycobacterium and it may be possible that some of the cross reactivity is due to this factor rather than the Vibrio harveyiVH1, moreover as the 32 kDa band highlighted is detected by all sera.  The cross reactivity of the VH1 vaccine is one of the important attributes of the selected vaccine and an important part of the paper.  Therefore, this needs to be tested again with pre-immunisation and adjuvant-only controls, and alongside again the Vibrio extracts to act as positive controls for the original results – and the results need to be shown.

-Pre-immunisation results were included in the figures. Besides, to support the results, supplementary data on the antigenically heterologous 32 kDa OMP of V. harveyi strain VH1 as a potential vaccine candidate was included. Please check line 332-346 and supplementary material.

“Overall, the western immunodetection using different Vibrios and Photobacterium damselae antisera revealed that V. harveyi strain VH1 antiserum induced the strongest antigenic response against the homologous and heterologous outer membrane protein antigens of the tested Vibrio spp. and Photobacterium damselae strains, indicated by the high intensity of the 32 kDa OMP bands. The strain was later used by Mursidi (unpublished) as a recombinant vhDnaJ vaccine by cloning the vhDnaJ gene encoding 32 kDa antigenic outer membrane protein (OMP) of V. harveyi strain VH1 with pET-32 Ek/LIC (Novagen, USA) vector confirmed by sequencing, analysed by bioinformatics analysis and followed with expression in a heterologous expression system, Escherichia coli BL21 (DE3) strain recombinant vaccine by cloning and sequencing the gene encoding 32 kDa antigenic outer membrane. Bioinformatics analysis revealed vhDnaJ gene from the 32 kDa antigenic OMP of V. harveyi strain VH1 was distributed widely and highly conserved in the Vibrio species as well as highly antigenic (Supplementary Material). Therefore, the V. harveyi strain VH1 was selected as an inactivated vaccine candidate for oral vaccine development in the current”.

 https://drive.google.com/file/d/1j0UXFcIr146lsA2fy-DuhH0lXVma5v1Y/view?usp=sharing

Reviewer 3 Report

Thanks for your changes.

However, most edit requested for figures and tables were not implemented, despite you replied "The figure/table was revised accordingly".

Author Response

Cover letter

On behalf of all the authors I, Aslah Mohamad states that there is no conflict of interest about the study submitted to the journal for possible publication.

Many thanks

Sequel to your mail to us on the review of our manuscript title “Laboratory and Field Assessments of oral Vibrio vaccine provides good protection against vibriosis in cultured marine fishes” the following correction and rebuttals were made:

Comments and suggestions from the reviewer (Reviewer 3)

All line numbers refer to the pdf version of the submission.

L 130: Vibrio strains

  • The word Vibrio was italic

L131: ultracentrifuged

  • The word was changed

L158: “Each of the rabbits” – there were four whole-cell protein preparations, so it must be two rabbits/Vibrio strain

  • Yes, true. Two rabbits/strain

L170: “All polyclonal rabbit antisera were incubated”. Rephrase, the membrane was incubated with…

  • The method was rephrased in Line 181

L170-176 are confusing. It appears that the membrane was incubated twice

  • The membrane was incubated once in blocking solution, once in blocking buffer with antiserum and finally once with HRP,

L188: “In this study, we used a specific strain”. It’s not clear why have you used this strain for. If your experimental method of the present paper is based on past observation please cite the reference

  • The word we used a specific strain was removed. The use of the strain from previous literature was mentioned in Line 102 – 113. “Our previous study on a feed-based whole-cell polyvalent vaccine against vibriosis, streptococcosis and motile aeromonad septicemia in Asian Seabass, Lates calcarifer showed that the oral polyvalent vaccine can provide around 75-80% protection after challenge with harveyi, A. hydrophila, and S. agalactiae ( Mohamad et al., 2021). The Vibrio harveyi strain VH1 vaccine which was a part of the polyvalent vaccine was still not tested as a single oral vaccine against different Vibrio spp. thus, was used in the current study to determine its ability to develop an antibody response shared against other Vibrio pathogens. Therefore, this research study proposes an oral Vibrio harveyi strain VH1 vaccine candidate that can provide good protection under laboratory, i.p. administrated, challenge models against three major Vibrio species; V. harveyi, V. parahaemolyticus, and V. alginolyticus, and can potentially improve antibody response, survival, and growth performance of farm marine fish”.

L196: “Bacterial cells of each strain were then inactivated”… Earlier you wrote that only strain VH1 was used for vaccine production, while here the sentence implies that all strain were grown and inactivated.

  • The words “Bacterial cells of each strain were then inactivated”… was changed to “In this study, only Formalin-Killed Cells (FKC) of harveyi strain VH1 prepared earlier was used for the feed vaccine preparation” in Line 199.

L204: It’s the first time I personally see palm oil being used as adjuvant. Authors should justify this choice and provide adequate reference for it.

  • The use of palm oil was further discussed in discussion part in Line 518-529.

L209: provide more details on the patent or at least the patent agency to which it was filed.

  • The patent agency (MyIPO, Malaysia) was included in the text in Line 210.

Fig1 is actually not very insightful. It might be improved by providing an overall graphical representation of the study rationale, with a summary of vaccinations, challenges and analysis performed on lab and field specimens per time point. Readers could benefit from a clearer view on the experimental procedure and source of bacterial material used (whole cell extract, OMP, FKC Vibrio strains)

  • Figure 1 was intent to give a general view on vaccination regime. We want to make it as simple as it can so that readers (which we target as farmers) can copy the regiment and apply. However, we agree with the reviewer to make it more insightful. Figure 1 was changed according to reviewer’s suggestion.

L212: University

  • The word Universiti is remain because it is a noun.

L213: Provide some sort of variability measure of the average weight

  • Variability measure was included in the text in Line 216.

L217: Control feed contained PBS and palm oil, not just PBS – you wrote this at line 206

- Yes, thank you for noticing this. The word “palm oil” was included in Line 220.  

L218: “Single and double boosters were given…”. How could these two vaccination strategies (one with a single booster, a second with two boosters) be tested if only one vaccination group (in duplicates) was used? Did each replicate received a single booster and a double booster?

  • thank you for noticing this. It should be first and second booster doses.

L254: How can you safely declare that no abnormal behaviour was associated to the challenge procedure if fish were anaesthetized in that moment?

  • The word was removed as all fish was inactive post anaesthetized.

L282: Were t test assumptions verified for all three datasets?

  • T-test assumptions was used for all datasets because we only have two groups to compare with and the data was distributed normally.

Fig3: I believe that Figure 3 doesn’t meet quality standards. Also, if I’m not wrong, SDS-PAGE or Western blot images are requested in their non-cropped original versions by all MDPI journals. Also, not all bands were reported in the text when describing obtained results or pointed by arrows. Why?

  • We only have that figure at the moment as the student doing the test already left the university. However, we would really appreciate if the reviewer can consider the figure. Only important bands were reported in the text.

Fig4: Same comment as above plus single images are misplaced in the panel and the meaning of black and red asterisks are not evincible from the legend. Also, figure legend should read “Immunoblot profiles of outer membrane proteins (OMPs) from four Vibrio spp and P. damselae….”

  • Figure 4 legend was improved. Nevertheless, we are sorry as for the figure, we only have this one.

Fig. 5: The first booster at week 2 and the second booster at week 6 aren’t properly indicated in the plot. Correspondence between IgM values/trends in the text sometime are not reflected in the plot as well.

  • The figure was improved. The IgM trends is mentioned in text.

L345: With respect and Asian

  • The typo was corrected

Table 2: 10 fish from vaccinated/non vaccinated groups were challenged in duplicates. This means 20 fish per group per challenge type. Colum 3 should therefore be amended. Values reported in column 4 don’t make any biological sense (how could 2.5 fish be dead?). I assume this is an average±variability (SD or SE). Not too sure about the values in the last two columns. If 20 and not 10 fish were challenged per group per challenge type, mortality cannot just be number of dead fish by 10.

  • Column 3 was improved and 4 was removed.

L362: Duration of field trial has always been expressed in weeks so far, please be consistent. Also, as is, it seems that fish were fed VH1-containing feed for the whole 4 weeks, which according to Fig. 2 is not the case.

  • Duration of vaccination trial was fixed as day

L366: I don’t get the possible relation of control/vaccinated feed and weight gain. Do authors attribute the weight gain exclusively to the vaccinated feed? Why is there such a discrepancy in the total amount of feed given to the two experimental groups? Were fish randomly sampled and weighed during the 16-week experiment to re-calibrate feed amounts to administer based on the 4% BW

  • The feed given was based on the weight and survival of the fish in every 14-days sampling. When the weight was increased in the vaccinated treatment, more feed was given. Yes, fish were randomly sampled every 2 weeks, measured their weight and length.

Fig.8: Please show all data points per each boxplot and report all details of the test statistics, not only the approximate p value. Also, the standard box plot visualization consists in the 25-75 quartile, aka Tukey in some statistical software, not min to max.

  • Figure 8 for boxplot was removed and changed to weight performance in every 14-days sampling.

Line 381: Did you employ percentage data directly in the t test analysis?

  • The data employed in the t test is based on the number of survived fish.

Fig.9: Normally asterisks are overlaid on the bar that represents the statistically significantly different group compared to the control.

  • The asterisks was overlaid on the bar

L388-389: Rephrase for clarity

  • The line was rephrased accordingly.

L413: It’s not clear to me why authors group keep on grouping P. damselae under the Vibrio spp. umbrella of species.

  • The line was improved. However Photobacterium damselae was previously known as Vibrio damsela and part of Vibrionaceae family which have been associated with health problems of marine animals similar to Vibrio parahaemolyticus, alginolyticus, V. harveyi, V. owensii, and V. campbellii (Ina-Salwany et al., 2019).

L435: In an attempt to justify ELISA results, the author state they expected systemic and mucosal IgM response to be stimulated. However, structural differences exist between mucosal and systemic IgM (polymer vs monomer) and no detail was given as to the ability of the Aquatic Diagnostics IgM monoclonal antibody employed to discriminate between the two. Systemic and mucosal IgM responses cannot be concluded.

  • The discussion was improved and the systemic and mucosal IgM responses was not concluded per suggestion by the reviewer.

L460: Again an uncertainty regarding weight increase. Authors justify that palm oil was responsible for weight increase. However, the control group received a feed supplemented with PBS AND palm oil as well. Their conclusion is not backed up by experimental design or results. Also, an excess of fatty acids may result in undesirable conditions i.e. dyslipidemia. Rather, were animal sacrificed at the end of the field survey and checked for any internal condition of the liver? Was the liver swollen?

  • The discussion was improved. We believe palm oil+vaccine can improve the growth performance rather than palm oil alone as we observed in few tests that being done with oral vaccines. However, we agree with the reviewer which we cannot conclude the palm oil as the growth promoter without sufficient supporting data. Thus, we concluded in the text, “Although Fraser et al. (2014) concluded that vaccination would reduce the growth of fish due to an increased regular metabolic rate following continuous stimulation of the immune system, this current study found that feeding the farm hybrid grouper with the oral harveyi strain VH1 vaccine can improve the growth performance of the fish. Amar et al. (2021) suggested that as fighting diseases and protecting against infections require a physiological cost, an ‘immune’ host could save energy for carrying and hosting pathogens, leaving more resources available for normal growth. Therefore, vaccination can promote growth by reducing the metabolic load of the immune response to infection”. In Line 513-520. The liver was not swollen. Only 10% palm oil was included in the vaccine formulation following (Aminudin et al., 2018; Firdaus-Nawi et al., 2013; Ismail et al., 2016; Monir et al., 2021).

Reference

Amar, E. C., Faisan, J. P., & Gapasin, R. S. J. (2021). Field efficacy evaluation of a formalin-inactivated white spot syndrome virus (WSSV) vaccine for the preventive management of WSSV infection in shrimp grow-out ponds. Aquaculture, 531(January 2018), 735907. https://doi.org/10.1016/j.aquaculture.2020.735907

Aminudin, A., Kamal, F. M., Zamri-saad, M., Abdullah, S., Ridzuan, M. S., Yusoff, H. M., Hashim, S., Sudirwan, F., Salihin, I., & Sulaiman, S. (2018). Effect of incorporating different concentrations of palm oil as adjuvant in fish vaccine. International Journal of Biosciences (IJB), 12(1), 35–41. https://doi.org/10.12692/ijb/12.1.35-41

Firdaus-Nawi, M., Yusoff, S. M., Yusof, H., Abdullah, S. Z., & Zamri-Saad, M. (2013). Efficacy of feed-based adjuvant vaccine against Streptococcus agalactiae in Oreochromis spp. in Malaysia. Aquaculture Research, 45(1), 87–96. https://doi.org/10.1111/j.1365-2109.2012.03207.x

Ina-Salwany, M. Y., Al-saari, N., Mohamad, A., Mursidi, F. A., Mohd-Aris, A., Amal, M. N. A., Kasai, H., Mino, S., Sawabe, T., & Zamri-Saad, M. (2019). Vibriosis in Fish: A Review on Disease Development and Prevention. Journal of Aquatic Animal Health, 31(1), 3–22. https://doi.org/10.1002/aah.10045

Ismail, M. S., Siti-Zahrah, A., Syafiq, M. R. M., Amal, M. N. A., Firdaus-Nawi, M., & Zamri-Saad, M. (2016). Feed-based vaccination regime against streptococcosis in red tilapia, Oreochromis niloticus x Oreochromis mossambicus. BMC Veterinary Research, 12(1), 1–6. https://doi.org/10.1186/s12917-016-0834-1

Mohamad, A., Zamri-Saad, M., Amal, M. N. A., Al-saari, N., Monir, M. S., Chin, Y. K., & Md Yasin, I.-S. (2021). Vaccine Efficacy of a Newly Developed Feed-Based Whole-Cell Polyvalent Vaccine against Vibriosis, Streptococcosis and Motile Aeromonad Septicemia in Asian Seabass, Lates calcarifer. Vaccines, 9(4), 368. https://doi.org/10.3390/vaccines9040368

Monir, M. S., Yusoff, M. S. M., Zulperi, Z. M., Hassim, H. A., Zamri-Saad, M., Amal, M. N. A., Salleh, A., Mohamad, A., Yie, L. J., & Ina-Salwany, M. Y. (2021). Immuno-protective efficiency of feed-based whole-cell inactivated bivalent vaccine against Streptococcus and Aeromonas infections in red hybrid tilapia (Oreochromis niloticus × Oreochromis mossambicus). Fish & Shellfish Immunology, 113, 162–175. https://doi.org/10.1016/j.fsi.2021.04.006

Round 3

Reviewer 2 Report

I have gone through the revised manuscript and the supplementary material in relation to my previous comments.

Unfortunately, I still cannot recommend the manuscript for publication in its current form.

The authors have addressed most of the points, have added details and have modified the text, tables and figures which results in a much improved manuscript, reflecting more accurately the findings and interpretation of the study.   

In my opinion the data presented on the cross antigenicity of the different Vibrio species are not robust due to the absence of a control for Freund’s Complete Adjuvant (FCA).  It cannot be said with certainty that the antibody binding to the 32kDa protein is Vibrio specific, and not generated against a shared epitope in the bacterial component of the FCA.

If I interpret the data correctly, then the supplementary data presented does not help in clarifying that the 32kDa Vibrio protein is antigenic.  Instead the western blot highlighting the 32kDa protein + His tag, uses an anti-histidine antibody – not primary antibodies generated against Vibrio extracts in rabbit or fish.

Additionally, even though I overlooked it in the previous resubmission, I think that the presentation of results from different gels placed together in Figure 4 is not robust.  Given variation in western blot staining then why were not all Vibrio species and the relevant pre-immune serum run on the same gel for testing with a given Vibrio anti-serum? For example, given that the pre-serum seems to have been run on a separate gel, then where is the evidence that the detection worked on that occasion, etc. etc.  Even the marker seems to have been taken from a different gel to all others.

Perhaps there is evidence that there is a 32kDa protein which may display cross antigenicity, and perhaps it is indeed the protein identified in the supplementary material, but this may be just one of many proteins, and the real value of this protein can only be assessed through challenge trials, and even then that does not show that it alone is responsible for the cross antigenicity seen.

There is also no follow up with P. damselae in the challenges, so this part of the cross-antigenicity is not followed through.  Moreover, V. harveyi has already been published as part of the polyvalent vaccine, so a choice on its antigenicity has already been made. 

Overall therefore, given the issues with the controls, in my opinion the work relating to the cross-antigenicity based on the rabbit sera, and the basis for selecting V. harveyi, could be removed.  The results on the cross antigenicity based on sera analysis from the lab challenges, the results of the lab and field challenges, plus the interesting finding on weight gain in vaccinated fish, is probably sufficient for publication – this combined with the already improved discussion and interpretation of the vaccine’s potential.

I would recommend to remove the section on basis for selection of V. harveyi (rabbit sera cross-antigenicity) and to reformat with the remaining results.  The manuscript would then be suitable for publication.

Author Response

Cover letter

On behalf of all the authors, I, Aslah Mohamad, states that there is no conflict of interest about the study submitted to the journal for possible publication.

Many thanks

Sequel to your mail to us on the review of our manuscript title “Laboratory and Field Assessments of oral Vibrio vaccine indicate the potential for protection against vibriosis in cultured marine fishes” the following correction and rebuttals were made:

Comments and suggestions from the reviewer (Reviewer 2-Third revision)

General comments

I would recommend to remove the section on basis for selection of V. harveyi (rabbit sera cross-antigenicity) and to reformat with the remaining results.  The manuscript would then be suitable for publication.

  • Thank you for your time and review on improving the manuscript. The section on the basis for the selection of  harveyi(rabbit sera cross-antigenicity) and its results was removed following the reviewer’s recommendation. Thank you again for your kind considerations.
